# Structural basis for kinase inhibition in the tripartite *E. coli* HipBST toxin–antitoxin system

René L Bærentsen[1†], Stine V Nielsen[2†], Ragnhild B Skjerning[1], Jeppe Lyngsø[3], Francesco Bisiak[1], Jan Skov Pedersen[3], Kenn Gerdes[4], Michael A Sørensen[2]*, Ditlev E Brodersen[1]*

[1]Department of Molecular Biology and Genetics, Aarhus University, Aarhus, Denmark; [2]Department of Biology, University of Copenhagen, Copenhagen, Denmark; [3]Department of Chemistry and Interdisciplinary Nanoscience Centre (iNANO), Aarhus, Denmark; [4]Voldmestergade, Copenhagen, Denmark

**Abstract** Many bacteria encode multiple toxin–antitoxin (TA) systems targeting separate, but closely related, cellular functions. The toxin of the *Escherichia coli hipBA* system, HipA, is a kinase that inhibits translation via phosphorylation of glutamyl-tRNA synthetase. Enteropathogenic *E. coli* O127:H6 encodes the *hipBA*-like, tripartite TA system; *hipBST*, in which the HipT toxin specifically targets the tryptophanyl-tRNA synthetase, TrpS. Notably, in the tripartite system, the function as antitoxin has been taken over by the third protein, HipS, but the molecular details of how activity of HipT is inhibited remain poorly understood. Here, we show that HipBST is structurally different from *E. coli* HipBA and that the unique HipS protein, which is homologous to the N-terminal subdomain of HipA, inhibits the kinase through insertion of a conserved Trp residue into the active site. We also show how auto-phosphorylation at two conserved sites in the kinase toxin serve different roles and affect the ability of HipS to neutralize HipT. Finally, solution structural studies show how phosphorylation affects overall TA complex flexibility.

*For correspondence:
mas@bio.ku.dk (MAS);
deb@mbg.au.dk (DEB)

†These authors contributed equally to this work

Competing interest: The authors declare that no competing interests exist.

## eLife assessment

This **important** study presents an exhaustive structural analysis of a complete tripartite HipBST toxin-antitoxin system of the Enteropathogenic *E. coli* O127:H6, which represents a fascinating variation on the well-studied HipAB toxin-antitoxin system. The **convincing** data show that major features of the canonical HipAB system have been rerouted to form the tripartite HipBST, revealing a new mode of inhibition of a toxin kinase.

## Introduction

Bacteria employ a wide range of mechanisms to adapt to changes in their environments and imminent threats, such as exposure to antibiotics and phages, including activation of toxin–antitoxin (TA) systems that encode intracellular toxins able to cause rapid growth adaptation (*LeRoux and Laub, 2022*). Canonical (type II) TA systems are small, bicistronic loci that encode a protein toxin and its cognate antidote (antitoxin) that tightly interact to form an inactive higher-order complex, usually also capable of controlling transcription via a DNA-binding domain on the antitoxin (*Harms et al., 2018*). However, there are also examples of tricistronic TA loci, in which the third gene either encodes a chaperone required for folding of the antitoxin and thus, toxin inhibition (*Bordes et al., 2016*) or

an additional, transcriptional regulator (*Hallez et al., 2010*; *Zielenkiewicz and Ceglowski, 2005*; *Jurėnas et al., 2021*).

In the widespread and diverse *hipBA* (*high persister*) system, the HipA toxin is a serine–threonine kinase (STK) (*Hanks et al., 1988*; *Stancik et al., 2018*), while the cognate antitoxin, HipB, contains a helix-turn-helix (HTH) motif (*Gerdes et al., 2021*; *Schumacher et al., 2009*). For the *hipBA* system from *Escherichia coli* K-12, the toxin (HipA$_{Ec}$) specifically targets glutamyl-tRNA synthetase (GltX) by phosphorylation of an ATP-binding motif conserved in type I aminoacyl-tRNA synthetases (*Eriani et al., 1990*; *Sekine et al., 2003*), thereby inhibiting its activity and blocking translation (*Germain et al., 2013*). Accumulation of uncharged tRNA$^{Glt}$ subsequently induces the stringent response and cellular dormancy via RelA-mediated (p)ppGpp synthesis on starved ribosomes (*Haseltine and Block, 1973*; *Pacios et al., 2020*; *Winther et al., 2018*). In contrast to most classical type II TA systems, the antitoxin, HipB, does not directly block the HipA active site so inhibition has been proposed to occur by several alternative mechanisms, including inhibition of conformational changes in the HipA kinase required for catalysis, sequestration of HipA on DNA via the HipB DNA-binding domain (*Schumacher et al., 2009*), and allosterically via placement of a C-terminal Trp residue of HipB into a pocket on HipA (*Evdokimov et al., 2009*). HipA$_{Ec}$ is also regulated by *trans* auto-phosphorylation at a conserved serine (Ser150) situated in a loop near the active site termed the 'Gly-rich loop' (residues 151–156), which is required for ATP binding and functionally (but not structurally) similar to the P loop of eukaryotic kinases (*Huse and Kuriyan, 2002*; *Schumacher et al., 2012*). Two discrete conformations of the Gly-rich loop are observed depending on the phosphorylation state of Ser150 and whether substrate (ATP) is bound or not (*Schumacher et al., 2015*; *Schumacher et al., 2012*; *Schumacher et al., 2009*). In the unphosphorylated state and in the presence of ATP, the flexible Gly-rich loop is found in an inward conformation in which main chain amino groups coordinate several ATP phosphate groups and the kinase is active (*Schumacher et al., 2009*). Phosphorylation of Ser150, on the other hand, results in strong interactions of the phosphate group with several active sites residues, which causes ejection of the Gly-rich loop and prevents binding of ATP, thus inactivating the kinase (*Schumacher et al., 2012*; *Wen et al., 2014*).

In addition to the canonical *hipBA* locus, enteropathogenic *E. coli* O127:H6 contains a partially homologous, tricistronic Hip system; *hipBST*, encoding three separate proteins. HipB, by analogy to HipBA, contains a putative DNA-binding HTH domain, suggesting a function as transcriptional regulator, while HipT is a small HipA-like kinase that specifically targets tryptophanyl-tRNA synthetase, TrpS (*Gerdes et al., 2021*; *Vang Nielsen et al., 2019*). Surprisingly, the third protein, HipS, which at the sequence level corresponds to the N-terminal subdomain 1 of HipA (the N-subdomain 1), was shown to function as the antitoxin of HipT but the molecular mechanism by which HipS neutralizes HipT is currently not known (*Vang Nielsen et al., 2019*). Phylogenetic analysis has demonstrated that HipT and HipA toxins are closely related within a very broad superfamily of HipA-homologous kinases (*Gerdes et al., 2021*). HipT has two phosphoserine positions, Ser57 and Ser59, in its Gly-rich loop (residues 58–63), which are both modified by *trans* auto-phosphorylation in vivo suggesting that they play a role in regulating the activity of HipT (*Vang Nielsen et al., 2019*).

Here, we determine a 2.9-Å crystal structure of a kinase-inactive HipBST$^{D233Q}$ complex, revealing that the overall structure is markedly different from *E. coli* HipBA but similar to a HipBA complex from *Shewanella oneidensis*. In this structure, HipT interacts tightly with HipS and that despite the lack of phosphorylation we find the Gly-rich loop in an outward, inactive confirmation similar to phosphorylated HipA, which we show is likely a result of insertion of a bulky Trp residue in HipS into the kinase active site. Next, we show that both of the serine residues in the Gly-loop are essential for HipT toxicity in vivo and that auto-phosphorylation of Ser59 prevents neutralization by HipS. We next determine crystal structures of two kinase active HipBST variants revealing that auto-phosphorylation takes place at both Ser57 and Ser59 despite the lack of in vivo toxicity. In contrast to *E. coli* HipA, phosphorylation of Ser57 or Ser59 does not affect the conformation of HipT inside the HipBST complex, which remains in an inactive state with the Gly-rich loop ejected, which we attribute to the presence of HipS. Finally, we show that the flexibility of the complex in solution varies in a phosphorylation-dependent manner and use a phylogenetic analysis to demonstrate that variations of the HipT Gly-rich loop serine residues correlate with separate clades of proteins across bacterial species.

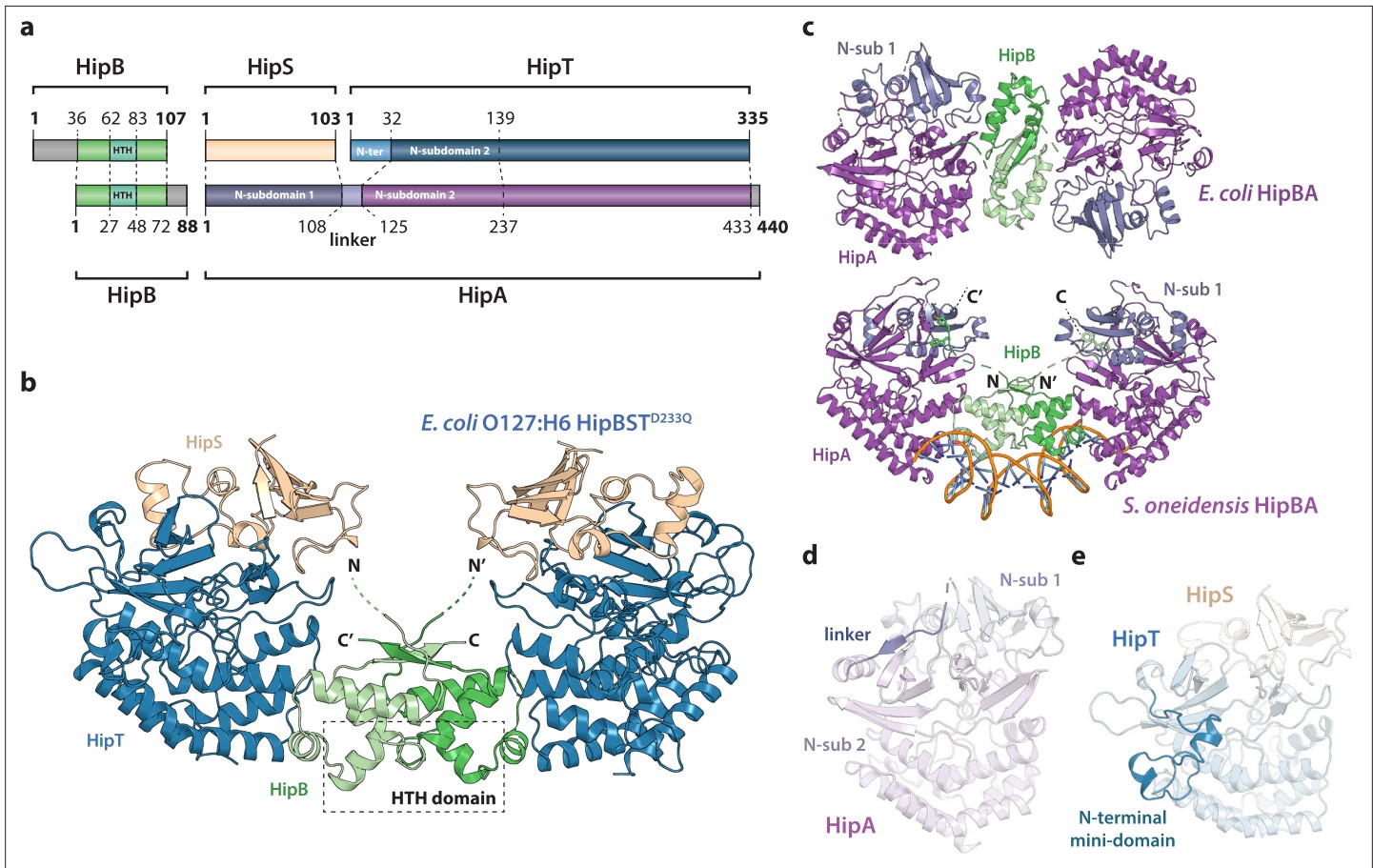

**Figure 1.** Crystal structure of the *E. coli* O127:H6 HipBST complex. (**a**) Schematic representation of HipBST and HipBA$_{Ec}$ showing corresponding proteins and domains (dashed lines) while grey areas represent regions missing in the crystal structures: *E. coli* O127:H6 HipBST, this work; *E. coli* HipBA, PDB: 2WIU (***Evdokimov et al., 2009***). HipS (beige) is structurally similar to the N-subdomain 1 of HipA (dark purple), HipT has an additional N-terminal mini-domain not found in HipA (light blue), and HipA has a short linker between the two N-subdomains (light purple). (**b**) Crystal structure of *E. coli* O127:H6 HipBST$^{D233Q}$ reveals a hetero-hexameric complex with HipS (beige) tightly associated with HipT (blue) and connected to a similar heterodimer through dimerization of HipB (green), which also generates a DNA-binding helix-turn-helix motif (HTH, dashed box). (**c**) Top, crystal structure of the *E. coli* K-12 HipBA complex (PDB: 2WIU) shown as cartoon with HipA in purple and blue (N-terminal subdomain 1) and the HipB homodimer in two shades of green (***Evdokimov et al., 2009***); bottom, the structure of *S. oneidensis* HipBA (PDB: 4PU3, HipA purple/blue, HipB green) bound to DNA (orange backbone, blue bases) (***Wen et al., 2014***). (**d**) The linker (dark purple) between N-subdomains 1 (N-sub 1) and 2 (N-sub 2) physically linking the two domains in HipA$_{Ec}$. (**e**) The N-terminal mini-domain of HipT (blue), which is absent from HipA.

The online version of this article includes the following source data and figure supplement(s) for figure 1:

**Figure supplement 1.** Sequence alignment with consensus elements of HipBST, HipBA$_{Ec}$ and HipBA$_{So}$.

**Figure supplement 2.** Transcriptional regulation of the HipBST promoter of *E. coli* O127:H6.

**Figure supplement 2—source data 1.** Original glucose and arabinose plates.

**Figure supplement 3.** Structural homologue analysis of HipBST to HipBA.

## Results

### *E. coli* HipBST forms an heterohexameric complex with HipT in a canonical inactive conformation

To understand the molecular and structural basis for the functional differences between the HipBA and HipBST TA systems, we determined the crystal structure of HipBST$^{D233Q}$, containing the inactive HipT D233Q kinase variant, to 2.9 Å by molecular replacement using HipA$_{Ec}$ as search model (***Figure 1a, b***, ***Supplementary file 1a***). The refined structure ($R$ = 19.5%, $R_{free}$ = 22.8%) has two copies of each of the three proteins in the asymmetric unit (ASU) and is generally well defined for HipB and HipT, while HipS is less well ordered. The higher-order HipBST complex structure consists of two HipST toxin:antitoxin

complexes separated by a dimer of HipB proteins. This differs markedly from the HipBA$_{Ec}$ complex, in which two HipA toxins (corresponding to HipT + HipS) are closely packed head-to-tail around two HipB antitoxins (**Figure 1c**, top) but is reminiscent of the structure of a HipBA complex from *S. oneidensis* (HipBA$_{So}$), which was crystallized in complex with DNA (**Figure 1c**, bottom) (**Wen et al., 2014**).

HipB consists of an α-helical bundle of four helices, including the central, DNA-binding HTH motif (residues 62–83), which is exposed at the bridge of the hetero-hexamer (**Figure 1b**, green). The helical bundle is followed by a small β strand that forms an antiparallel sheet by pairing with the corresponding stretch in the neighbouring HipB protein, thus forming a homodimer. The N-terminus of HipB (residues 1–35) is not visible in the electron density, but AlphaFold2 (**Jumper et al., 2021**) prediction indicates that it forms a single α-helix (**Figure 1—figure supplement 1a**). For HipBA$_{Ec}$, a hydrophobic residue at the C-terminus of HipB was proposed to reach a cleft between the N-subdomain 1 and the core kinase domain of HipA and inhibit activity by restricting domain movements required for catalysis (**Figure 1c**; **Schumacher et al., 2009**). HipB from HipBST does not have such a C-terminal residue (**Figure 1—figure supplement 1a**), supporting that HipB functions as a transcriptional regulator rather than the antitoxin. To confirm this, we constructed a vector-based transcriptional fusion reporter, in which the native *hipBST* promoter region and 5′ region of the *hipB* gene were transcriptionally fused to *lacZYA* (**Figure 1—figure supplement 2**, top) and confirmed that the promoter was active (**Figure 1—figure supplement 2**). Expression of HipBST$^{D233Q}$ or HipB alone efficiently repressed transcription, while expression of either HipS or HipT$^{D233Q}$ alone did not affect transcription. Thus, *hipB* is required, and sufficient, for transcriptional repression of the *hipBST* promoter. With respect to DNA binding, superposition of HipBST with HipBA$_{So}$ in complex with DNA (**Figure 1—figure supplement 3a**) shows that HipT has several conserved, positively charged residues near the expected location of the DNA phosphodiester backbone, supporting that HipBST binds DNA in a similar way to HipBA$_{So}$. This is also consistent with the observation that even a very low, leaky expression of *hipBST* or *hipBT* were sufficient to reduce transcription (**Figure 1—figure supplement 2**, left plate), while *hipB* alone needed to be expressed at high levels to repress the *hipBST* promoter.

HipS interacts tightly with HipT in the HipBST complex and consists of a solvent-facing, five-stranded anti-parallel β sheet of which four strands are located in the N terminus (residues 1–52). A small domain of three helices (residues 53–94) forms the interface to HipT and is followed by the fifth β strand (residues 95–102, **Figure 1b**). As expected from sequence analysis (**Figure 1—figure supplement 1b**), the fold of the HipST heterodimer is structurally very similar to the longer HipA kinase, such as HipA$_{Ec}$ (**Figure 1—figure supplement 3b**). In other words, HipS is structurally homologous to the N-subdomain 1 of HipA (**Schumacher et al., 2012**), both with respect to its overall fold and to its orientation with respect to the core kinase domain (**Figure 1—figure supplement 3b**). However, the HipS C-terminus overlaps with an extended linker that bridges N-subdomains 1 and 2 in HipA (**Figure 1d** and **Figure 1—figure supplement 3b**). HipT, on its part, corresponds structurally to the core kinase domain of HipA, except for an additional N-terminal mini-domain of unknown significance (residues 1–41, **Figure 1a, e**). Finally, in this structure, the Gly-rich loop of HipT is found in an outward-facing conformation despite the absence of Gly-rich loop phosphorylation, suggesting that HipT is inactive (**Figure 1—figure supplement 3d**).

In conclusion, we find that the overall structure of the tripartite HipBST complex is markedly different from the canonical *E. coli* HipBA complex, but structurally similar to a HipBA complex from *S. oneidensis*. Key features of HipB proposed to be involved in the kinase toxin inhibition mechanism in both HipBA$_{Ec}$ and HipBA$_{So}$ are missing. Finally, HipT appears in a canonical inactive conformation despite being unphosphorylated, suggesting a different mechanism of toxin inactivation and inhibition.

## HipS inhibits HipT through insertion of a hydrophobic residue near the active site

The observation that HipT is found in a canonical, inactive conformation despite being unphosphorylated led us to hypothesize that HipS might neutralize HipT by direct interaction as seen in most other type II TA systems (**Harms et al., 2018**). This would also be consistent with the observation that HipS functionally acts as the antitoxin of the HipBST system in vivo (**Vang Nielsen et al., 2019**). There are three major areas of contact between the two proteins in the HipBST complex, involving both hydrogen bonds, hydrophobic and charged interactions (**Figure 2a**). Importantly, HipT and HipS

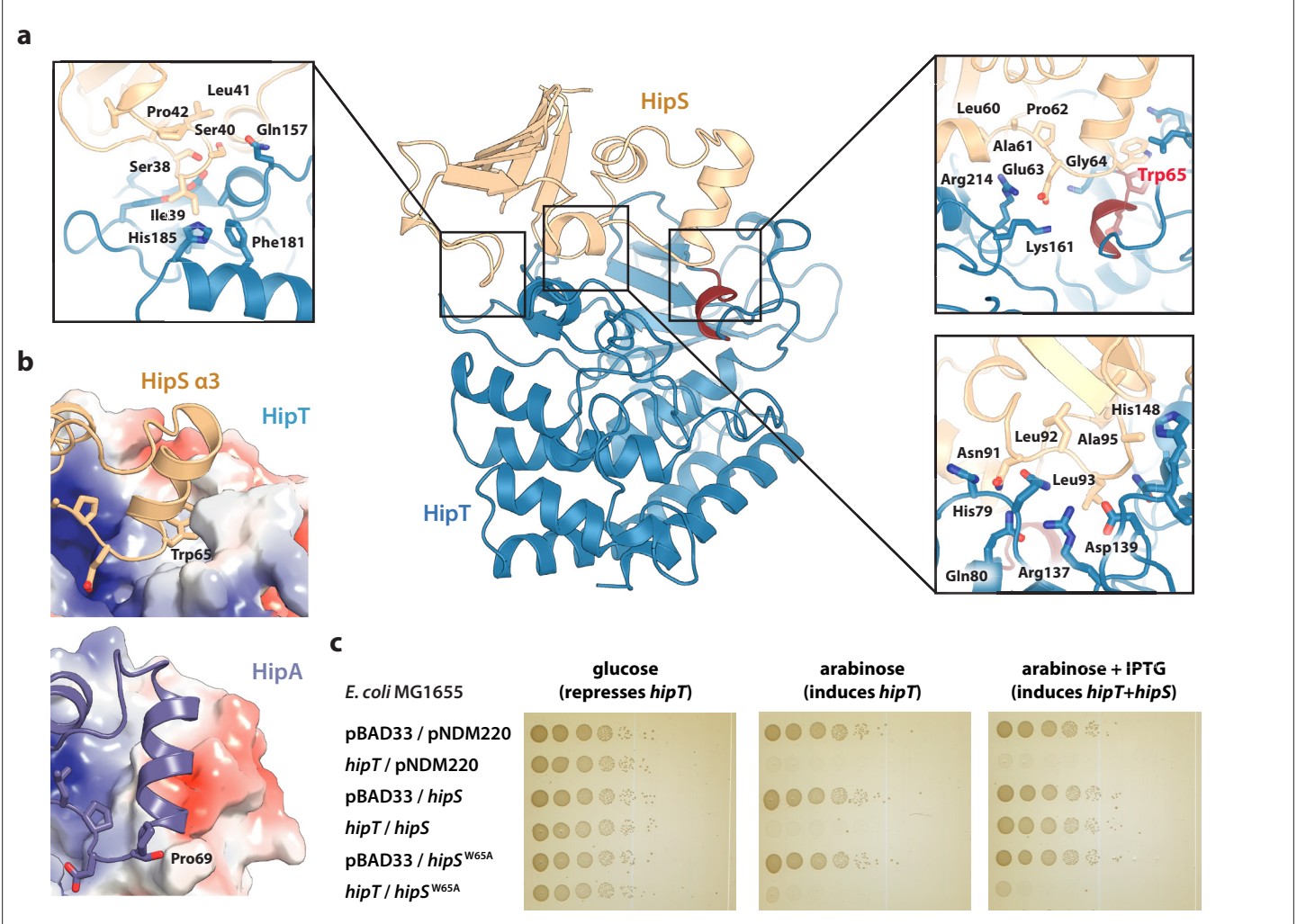

**Figure 2.** Trp65 is essential for the function of HipS as antitoxin. (**a**) Overview and detailed interactions between HipS (beige) and HipT (blue) at the three main areas of interaction with relevant residues indicated. The Gly-rich loop, including Trp65, is shown in red. (**b**) Top, HipS Trp65 (beige) is located in a hydrophobic pocket on the surface of HipT (coloured by surface electrostatics with APBS); bottom, this region is structurally different in *E. coli* HipA (purple and surface electrostatics shown, PDB: 3TPD) (*Schumacher et al., 2012*). (**c**) *E. coli* MG1655 harbouring empty pBAD33 vector (pBAD33) or pSVN1 (pBAD33::*hipT*) in combination with empty pNDM220 vector (pNDM220), pSVN109 (pNDM220::*hipS*), or pSVN178 (pNDM220::*hipS*^W65A), as indicated. Plates contained 0.2% glucose (to repress *hipT*), 0.2% arabinose (to induce *hipT*), or 0.2% arabinose plus 200 µM isopropyl-D-1 thio-galactopyranoside (IPTG) (to induce *hipS*, or *hipS*^W65A). The experiments shown are representative of at least three independent repetitions.

The online version of this article includes the following source data and figure supplement(s) for figure 2:

**Source data 1.** Original plates.

**Figure supplement 1.** Sequence and function of *E. coli* HipS.

were found to interact directly very close to the Gly-rich loop, where Glu63 from HipS points directly into the HipT active site while HipS Trp65 is wedged into a deep cavity formed by the Gly-rich loop in its outward conformation (*Figure 2a*, right). Of these, Glu63 is conserved in the N-subdomain 1 of the longer HipA toxins (*Figure 1—figure supplement 1b*), and therefore less likely contribute to functional differences between the two systems. Trp65, on the other hand, is found in many HipS orthologues (*Figure 2—figure supplement 1a*) and moreover, the region surrounding it is both structurally and electrostatically different from the corresponding region in HipA$_{Ec}$ (*Figure 2b*). Based on this, we hypothesized that insertion of the bulky Trp residue near the HipT active site might sterically block transition of the Gly-rich loop from its outward to the inward conformation, likely required for ATP binding and kinase activity. To test this, we generated a HipS^W65A variant and probed its ability to neutralize HipT toxicity upon ectopic expression of the two proteins using a two-plasmid system in *E.*

*coli* MG1655 (*Blattner et al., 1997*). Dilution spot assays confirmed that *wt* HipS, but not HipS$^{W65A}$, can neutralize toxicity of HipT in vivo (*Figure 2c*). This result was reproduced by growth assays in liquid culture (*Figure 2—figure supplement 1b*). In conclusion, we show that HipS interacts directly with the HipT active site and neutralizes the kinase by insertion of a conserved tryptophan residue, likely preventing structural transition from the inactive to the active conformation. This mode of inhibition where the toxin is sterically and physically blocked by the antitoxin thus closely resembles that observed in other TA systems.

## The two phosphoserine positions in HipT are important for toxicity

To understand the functional implications of the two phosphoserine positions in HipT, Ser57 and Ser59 (*Vang Nielsen et al., 2019*), we initially decided to investigate their role in toxicity. For this, we designed a set of HipT variants with the serine residues individually substituted for either alanine (A, phosphoablative) to prevent auto-phosphorylation, or aspartic acid (D, phosphomimetic) to mimic the phosphorylated form. Growth assays in liquid culture showed that both wild-type HipT (S$^{57}$IS$^{59}$; SIS), as well as HipT D$^{57}$IS$^{59}$ (DIS) and S$^{57}$ID$^{59}$ (SID), caused growth inhibition upon overexpression in *E. coli* MG1655 (*Figure 3a*), demonstrating that the phosphomimetic mutations do not prevent toxicity, and consequently, kinase activity of HipT towards its target. Moreover, since it has previously been shown that HipT is only observed with either Ser57 or Ser59 phosphorylated, it is unlikely that the effects are due to auto-phosphorylation of the remaining serine residue in either case (*Vang Nielsen et al., 2019*). In addition, while co-expression of HipS neutralized the toxicity of HipT SIS and SID variants, the antitoxin was unable to neutralize the toxicity of the HipT DIS variant, suggesting that Ser57, but not Ser59, negatively impacts the function of HipS as antitoxin. Finally, alanine substitutions at either Ser position almost (S$^{57}$IA$^{59}$; SIA) or completely (A$^{57}$IS$^{59}$; AIS) abolished HipT toxicity (*Figure 3a*). Thus, both serine residues (in either phosphorylation state) are important for HipT toxicity. In summary, we find that phosphorylation of Ser57 negatively affects the function of HipS as antitoxin under conditions where the HipT kinase is toxic. This represents a clear functional departure from HipA, where phosphorylation of Ser150 inactivates the toxin (*Schumacher et al., 2012*).

## The active site conformation of HipT is independent of phosphorylation

To understand how phosphorylation of the two serine residues affects the structure of the HipBST complex, we determined two additional crystal structures of the HipBST complex in context of the HipT AIS (2.4 Å) and SIA (3.4 Å) variants (*Supplementary file 1a*), which are non-toxic in vivo, but in principle maintain an intact kinase active site. Inspection of the difference electron density maps in the vicinity of the active site revealed that in both cases, additional density was present close to the non-mutated serine position (*Figure 3—figure supplement 1*). For the HipBST AIS variant, we found a strong peak corresponding to phosphorylation of Ser59 in both copies of HipT in the crystallographic ASU (*Figure 3—figure supplement 1a*), while for the HipBST SIA variant, phosphorylation of Ser57 was present but incomplete and could only be confidently modelled in one of the two HipT molecules (*Figure 3—figure supplement 1b*). In its phosphorylated state, Ser59 (P-Ser59) forms strong interactions to Lys161, His212, as well as to the catalytic Asp210 inside the active site (*Figure 3b*, top), while P-Ser57 is located further away and forms hydrogen bonds to Tyr162 and Asp210 (*Figure 3b*, bottom). Nevertheless, the Gly-rich loop maintains an outward conformation in both structures similar to the one observed before, likely incompatible with ATP binding. Taken together, the structures suggest that HipT is inactive in the context of the HipBST complex regardless of the phosphorylation state of the serine residues in the Gly-rich loop. This is distinctly different from HipA, where phosphorylation of Ser150 was found to convert the kinase from an active to an inactive form by causing ejection of the Gly-rich loop (*Schumacher et al., 2012*). To support our interpretation of the electron densities, we analysed the purified HipBST SIA and AIS variants on Coomassie-stained Phos-tag sodium dodecyl sulfate–polyacrylamide gel electrophoresis (SDS–PAGE) gels, which can separate different phosphoprotein species, using the inactive HipT$^{D210A}$ variant as control for unphosphorylated HipT. This experiment was carried out both with and without separation on a Heparin column which we found was able to separate complexes based on phosphorylation state. This experiment confirmed that HipT AIS primarily is found on the phosphorylated form (*Figure 3c*, upper band), while HipT SIA is mostly on the unphosphorylated form (*Figure 3c*, lower band), thus in both cases consistent with the structural analysis.

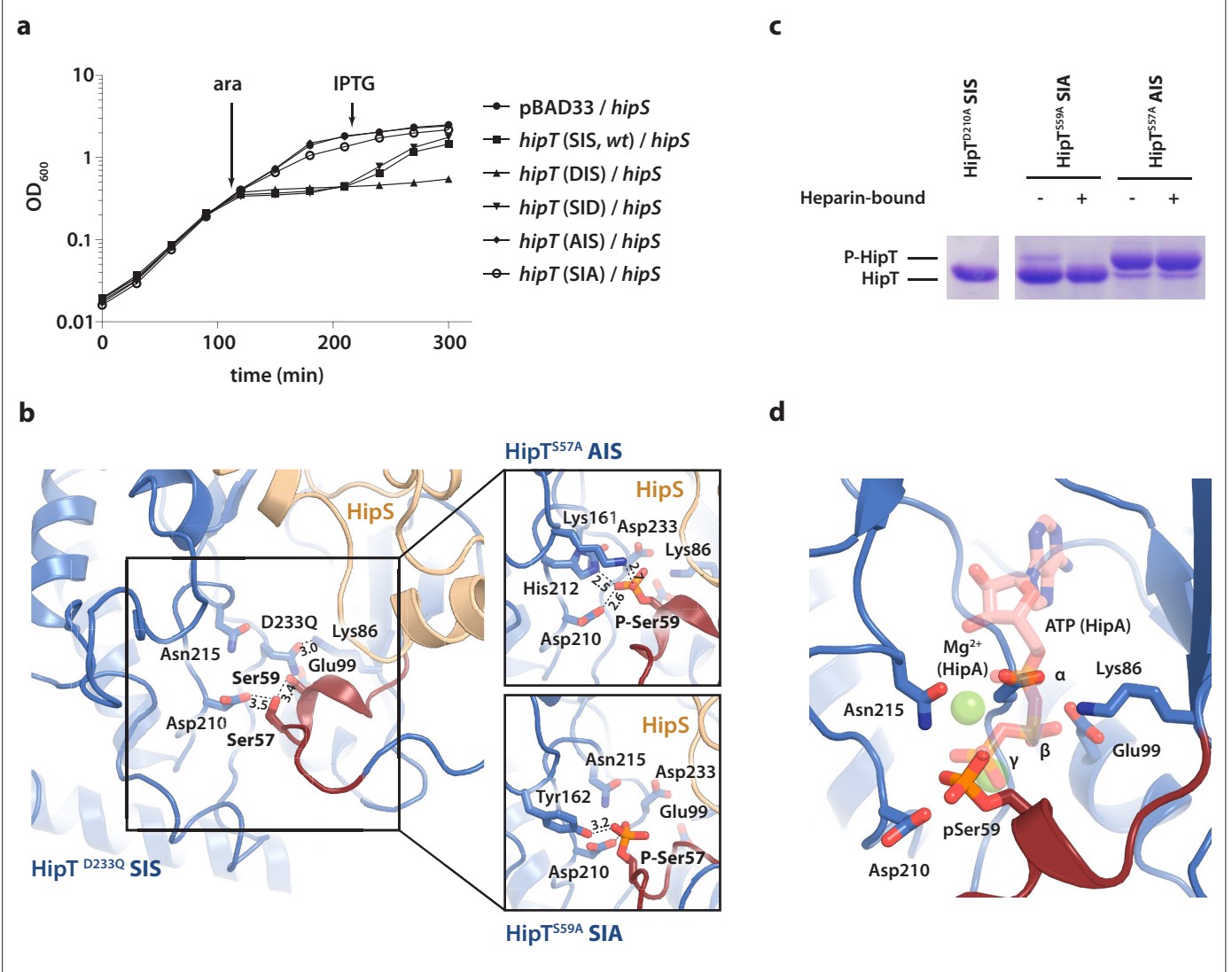

**Figure 3.** The phosphoserine positions in HipT have distinct functional roles. (**a**) Growth curves of *E. coli* MG1655 harbouring arabinose-inducible, single auto-phosphorylation variants of HipT; pBAD33::*hipT* (SIS, wt), pBAD33::*hipT*[S57D] (DIS), pBAD33::*hipT*[S59D] (SID), pBAD33::*hipT*[S57A] (AIS), pBAD33::*hipT*[S59A] (SIA), or the empty pBAD33 vector, co-transformed with an isopropyl-D-1 thio-galactopyranoside (IPTG)-inducible construct of HipS; pNDM220::*hipS*, with expression induced at the indicated time points (ara/IPTG). The curves show mean $OD_{600}$ values from at least two independent experiments with error bars indicating standard deviations (hidden when small). (**b**) Overview of the HipT kinase active site in the D233Q mutant as well as for HipT[S57A] (AIS, top) and HipT[S59A] (SIA, bottom). The phosphate groups on Ser57 (in HipT[S59A]) and Ser59 (in HipT[S57A]) are shown in orange and interacting nearby residues are highlighted. Numbers indicate distances in Å. (**c**) HipT variants from purified HipBST complex before (−) and after (+) a Heparin-column purification step aiming to separate complexes based on the phosphorylation state, visualized on a Phos-tag gel, which separates proteins based on phosphorylation state, and stained by Coomassie Blue. The locations of phosphorylated (P-HipT) and unphosphorylated (HipT) protein species are indicated. The gels are representative of two independent experiments. (**d**) Close-up of the HipT S57A active site overlaid with ATP (salmon, semi-transparent) and two $Mg^{2+}$ ions (green, semi-transparent) from the structure of HipA:ATP (PDB: 3DNT) (*Schumacher et al., 2009*). Relevant active site residues are shown as sticks and the Gly-rich loop is shown in dark red with pSer59 indicated.

The online version of this article includes the following source data and figure supplement(s) for figure 3:

**Source data 1.** SDS-PAGE.

**Figure supplement 1.** Electron density surrounding the phosphoserine sites of HipT before and after modelling.

In summary, we conclude that phosphorylation of neither of the two phosphoserine positions in HipT affects the conformation of the Gly-rich loop in the context of the HipBST complex, where the loop remains in an outward conformation in all cases suggesting the kinase is not in its canonical, activated state. This supports that phosphorylation itself is not responsible for inactivation of the kinase

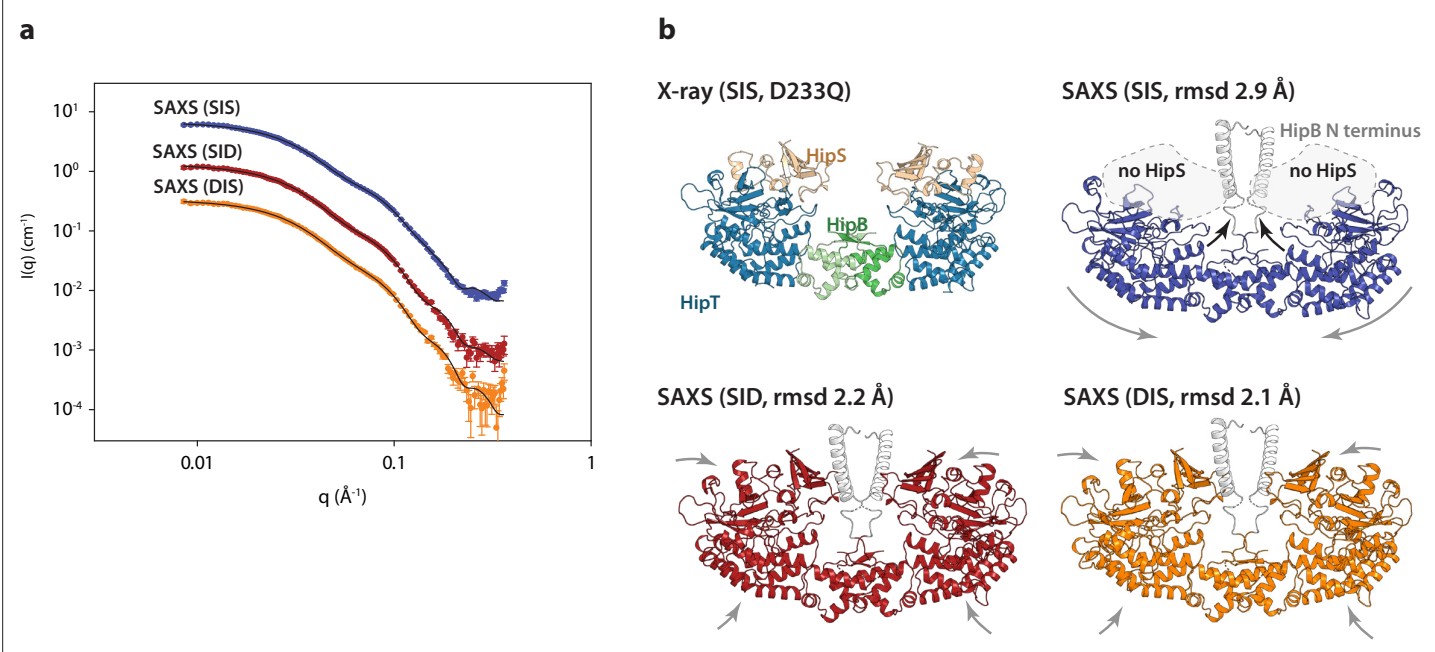

**Figure 4.** The HipBST complex is dynamic in solution. (**a**) Experimental small-angle X-ray scattering (SAXS) curves (measured X-ray intensity as a function of $q$, the modulus of the scattering vector) measured for HipBST in the context of HipT D210Q and the variants SIS (blue), SID (red), and DIS (orange). The black curves are the fits of the models based on the crystal structure after rigid-body refinement and inclusion of a structure factor to account for oligomerization of the complexes. (**b**) Structure models of HipBST (for SID, red, and DIS, orange) or HipBT (SIS, blue) as fit to the SAXS scattering data. For each model (SIS, SID, and DIS), the root mean square deviation (rmsd) to the crystal structure is indicated while arrows indicate gross domain movements. Top left, the crystal structure of the HipBST D233Q for reference.

The online version of this article includes the following source data and figure supplement(s) for figure 4:

**Figure supplement 1.** Small-angle X-ray scattering (SAXS) analysis of HipBST variants.

**Figure supplement 1—source data 1.** SDS-PAGE.

and is consistent with a specific role for HipS as an antitoxin. Surprisingly, auto-phosphorylation still takes place in both the SIA and AIS variants demonstrating that the kinase is still active and consequently, that the lack of toxicity observed for these variants likely relates to target binding rather than kinase inactivation. Finally, we note that the position of the Ser59 phosphate group in the AIS variant (HipT[S57A]) corresponds closely to that of the gamma-phosphate of ATP inside the active site of HipA, suggesting that phosphorylation in fact may stabilize the active site in a similar way to ATP (***Figure 3d***).

## HipBST dynamics in solution is affected by the phosphorylation state

To understand if the two phosphoserines affect the overall stability of HipBST, we purified complexes harbouring three different, phosphomimetic states of the Gly-rich loop (SIS, DIS, and SID) in the context of a kinase-inactive variant (HipT[D210A]) using a C-terminal His-tag on HipT. Intriguingly, for the protein purified with the wild-type SIS Gly-rich loop, the band corresponding to HipS was missing (***Figure 4—figure supplement 1a***). The outward state of the Gly-rich loop appears to be stabilized by an interaction between Asp210 and Ser59 and in a conformation compatible with HipS binding. Analysis of the purified samples of the three variants by small-angle X-ray scattering (SAXS) gave very similar intensity curves suggesting that the complexes have similar sizes and shapes in solution, despite the missing HipS protein in the SIS variant (***Figure 4a***). Moreover, all Guinier plots show a linear behaviour at low angles demonstrating the absence of large aggregates (***Figure 4—figure supplement 1b***) while the indirect Fourier transformation (***Figure 4—figure supplement 1c***) and pair distance distribution functions (***Figure 4—figure supplement 1d***) support similar, overall structures. Interestingly, the maximum diameter of 200 Å is significantly larger than the crystal structure (max. diameter 135 Å), consistent with analysis of the forward scattering that revealed partial oligomerization of the samples with an average mass corresponding to roughly a dimer of the HipBST

heterohexamer. Finally, the normalized Kratky plot agrees with a relatively compact structure without significant random coil content (*Figure 4—figure supplement 1e*).

We next attempted to model the SAXS data using the crystal structure of HipBST, by including a structural prediction of the missing HipB N-terminus and a hydration layer (*Figure 4*). As expected from the predicted oligomerization, this model was in relatively poor agreement with the measured data, so we allowed individual domains of the complex to undergo rigid-body motion while imposing C2 symmetry and at the same time refining the degree of oligomerization (*Steiner et al., 2018*). For each data set, the model with the best fit (lowest reduced $\chi^2$) to the data out of 10 runs was selected as the representative model. These three models (*Figure 4b*) all gave good fits to the data (*Figure 4a*) with $\chi^2$ (chi squared) values of, respectively, 2.2, 2.4, and 4.2 for SID, DIS and SIS, with varying degrees of oligomerization from 1.7 to 1.8 and an average distance between the heterohexamers ~80 Å. Consistent with the biochemical data, the SAXS data from the HipT SIS variant were matched very well by a model in which HipS was omitted and the disordered N-terminus of HipB was included as a helical element (*Figure 4*, blue). In this structure, the two HipT modules have rotated slightly to open up the cleft between them as compared to the crystal structure (*Figure 4b*, blue, arrows). On the contrary, data measured for both the SID (*Figure 4a, b*, red) and DIS (*Figure 4a, b*, orange) variants both matched the complete HipBST crystal structure (with the N-terminal HipB extension) with only minor adjustments of the domains. Taken together, SAXS analysis suggests that the HipBST complex is dynamic in solution in a phosphoserine-dependent way and that the phosphorylation state of HipT influences the inhibitory function of HipS. This dynamic behaviour likely explains how it is possible for the inactivated HipBST TA complex to undergo auto-phosphorylation in vivo.

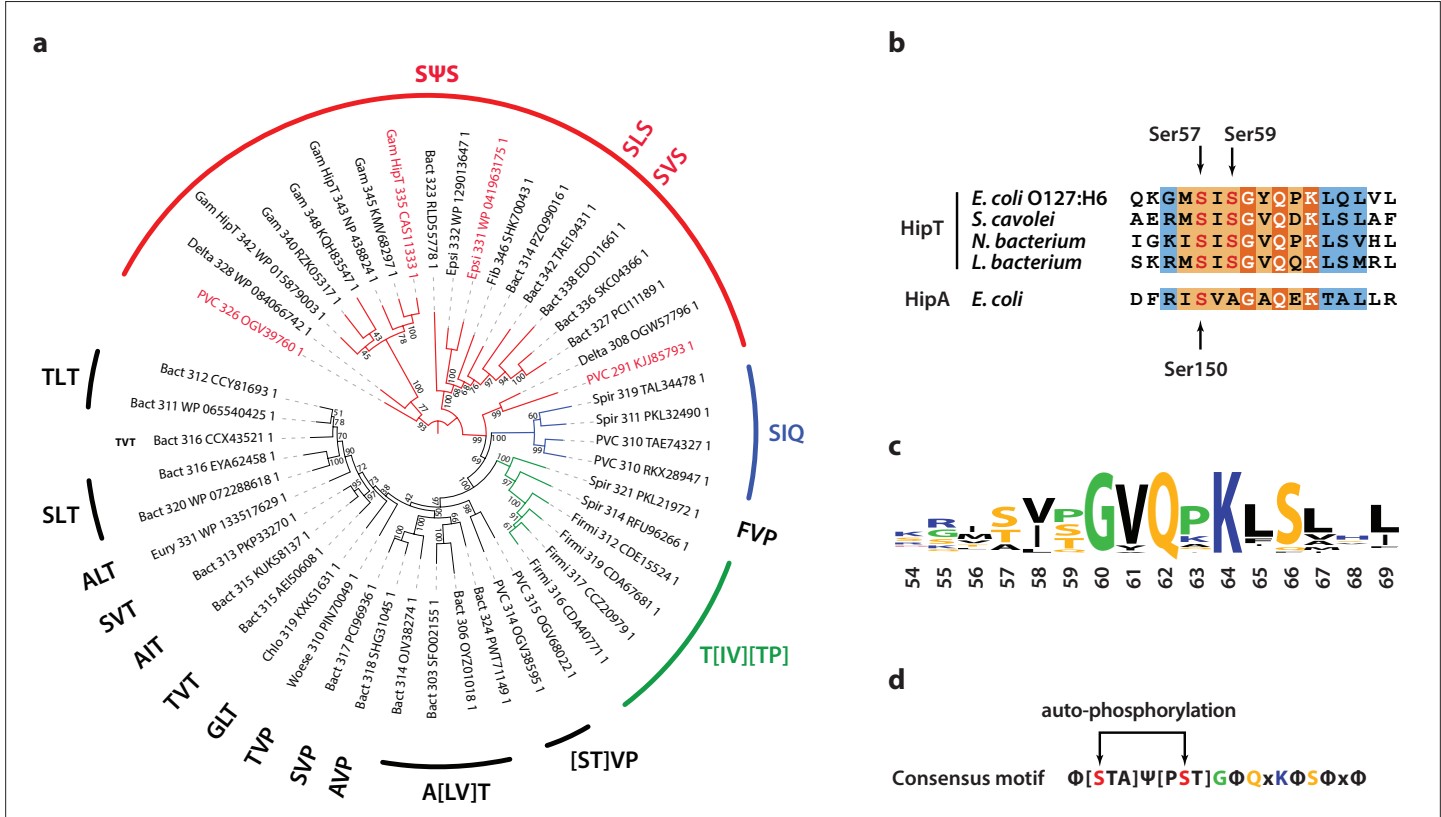

**Figure 5.** Phylogenetic analysis of HipT. (**a**) Phylogenetic guide tree of 48 HipT orthologues with sequences motifs (potential phosphorylation sites) indicated on the side. The SΨS group (red) is by far the largest group followed by the T[IV][TP] group (green), and SIQ group (blue). Sequences used in the alignment in (**b**) are shown with red letters and the numbers on the tree branches indicate bootstrap confidence levels. (**b**) Sequences of the Gly-rich loop (orange background), including the potential phosphorylation sites for selected HipT orthologues compared to HipA from *E. coli* K-12. Known phosphorylation sites in *E. coli* O127:H6 HipT (top) and *E. coli* K-12 HipA (bottom) are indicated with arrows and conserved sequence motifs with bold white text. (**c**) Sequence logo for the Gly-rich loop derived from all 48 HipT sequences. (**d**) Consensus motif for the HipT Gly-rich loop with known phosphorylation sites in red. Φ indicates a hydrophobic residue, while Ψ are aliphatic residues.

## The dual auto-phosphorylation sites are conserved among HipT kinases

Finally, we wanted to understand if Gly-rich loops with two phosphoserine positions are common among HipT-like kinases. For this, we generated a phylogenetic tree of the 48 known HipT orthologues in bacterial genomes (*Gerdes et al., 2021*; *Figure 5a*). Interestingly, investigation of this tree revealed that it largely separates the kinases by the configuration of the Gly-rich loop. The largest branch, which we named the S$\Psi$S group, contains HipT kinases with two serine residues separated by a small aliphatic amino acid, corresponding to the known auto-phosphorylation sites in HipT of *E. coli* O127:H6, Ser57 and Ser59 (*Figure 5a*). In many of the remaining HipT orthologues, this region contains various Ser/Thr motifs, including (S/T)$\Psi$T, (S/T)$\Psi$P, S$\Psi$Q, and $\Psi\Psi$T, but invariably with a central, small aliphatic residue ($\Psi$: Ile, Leu, or Val). Overall, the second Ser/Thr position appears less conserved than the first, and some kinases even contain a proline (P) or glutamine (Q) here, suggesting different regulation patterns for different HipT kinases with Ser57 playing a more important role. This supports the functional analysis showing that Ser57 affects neutralization by HipS in vivo and is less prone to auto-phosphorylation in the SIA variant. The remaining sequence of the Gly-rich loop across a range of HipT kinases is highly similar to HipA$_{Ec}$ (*Figure 5b, c*). Together, this allowed us to generate a consensus motif for the HipT Gly-rich loop, which includes several conserved hydrophobic positions in addition to the two Ser/Thr residues (*Figure 5d*). In summary, phylogenetic analysis reveals that a large fraction of HipT kinases contain a conserved S$\Psi$S motif with two potential phosphorylation positions in their Gly-rich loops, supporting that both Ser59, and in particular Ser57, play an important role in the regulation of the HipBST TA system.

## Discussion

In this study, we present a detailed structural and functional analysis of the tripartite HipBST TA system from enteropathogenic *E. coli* O127:H6, which elucidates the neutralizing mechanism of the unusual HipS antitoxin and dissects the functional roles of the two phosphoserine residues in HipT, Ser57 and Ser59. The crystal structure of the kinase-inactive HipBST$^{D233Q}$ complex shows that HipS interacts in a similar way to the corresponding N-subdomain 1 in HipA but neutralizes HipT by direction interaction involving insertion a large, bulky residue, Trp65, into the kinase active site. Trp65 is conserved in some, but not all HipS proteins, and is located in a motif that differs both structurally and at the sequence level from the corresponding region in HipA, which is otherwise very similar to HipS (*Gerdes et al., 2021*). Using multiple experimental approaches, we go on to show that both phosphomimetic and phosphoablative states of HipT impact toxicity and the interaction with the antitoxin HipS. Together, this suggests that the structural basis for the role of HipS as antitoxin in the HipBST system involves preventing the Gly-rich loop from transitioning from the outward, inactive state, to a conformation compatible with ATP binding. In some organisms, the residue corresponding to Trp65 is a proline, which may point to yet another way of preventing activation (*Figure 2—figure supplement 1a*).

Recently, structures of a HipT orthologue from *Legionella pneumophila* (HipT$_{Lp}$) in complex with either the non-hydrolyzable ATP analogue (AMPPNP) or the equivalent of *E. coli* HipS, HipS$_{Lp}$ as well as the isolated structure of a HipS from *Haemophilus influenzae* have been determined (*Lin et al., 2023*; *Zhen et al., 2022*; *Koo et al., 2022*). HipT$_{Ec}$ and HipT$_{Lp}$ differ significantly in the structure and size of the loops that were proposed to engage in target binding (*Gerdes et al., 2021*) suggesting that the two Hip kinases might have different targets (*Figure 6—figure supplement 1*). Moreover, the N-terminal minidomain that stretches from the N-terminus to the start of the Gly-rich loop is unresolved in the HipST$_{Lp}$ structure whilst being visible in the isolated structure of the HipT$_{Lp}$ toxin. This region is ordered in the structure of *E. coli* HipBST, which also suggests that the two kinases differ. HipT$_{Lp}$ belongs to the S$\Psi$Q group, and phosphorylation of the Gly-rich loop can thus take place only at the position corresponding to Ser57 in HipT$_{Ec}$. This allows us to further dissect the functions of the two phosphorylation sites in HipT$_{Ec}$. Interestingly, and despite being phosphorylated, HipT$_{Lp}$ is found with its active site loop in an inward and active conformation with nucleotide bound. In this conformation, the phosphoserine moiety is stabilized by two conserved arginine residues in a RxDR motif, which is found both in HipT$_{Ec}$ and HipA$_{Ec}$ (*Figure 1—figure supplement 1*). In HipA, however, the Gly-rich loop does not contact the RxDR motif in either phosphorylation state, suggesting that this interaction is a unique feature of HipT kinases (*Schumacher et al., 2012*). Gln56 (S$\Psi$**Q**), which corresponds to Ser59 in HipT$_{Ec}$ and arguably in some respects mimics a phosphorylated serine, interacts

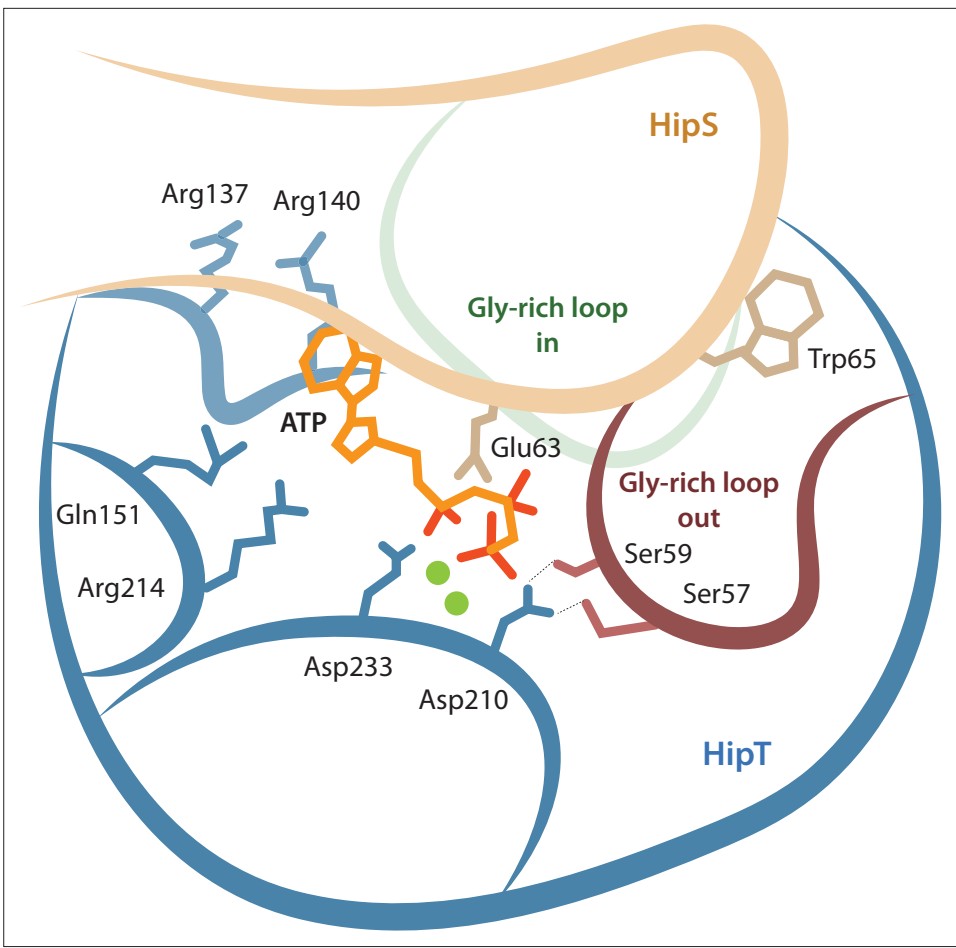

**Figure 6.** Model for the active site network of HipT of the HipBST system. Schematical overview of HipT showing the interactions found in this study. HipT (blue) with the observed outward conformation of the Gly-rich loop (red), and the predicted inward conformation (green). Important residues highlighted are Ser57, Ser59, Asp210, and Asp233 of HipT, and Trp65 of HipS (beige). Shown is also the position of an ATP molecule (orange) together with two Mg-ions (green spheres) based loosely on PDB: 7WCF. The two sites homologous to the sites found in HipT from *L. pneumophila* to interact with the phosphoserine positions (R137/R140 and Q151/R214 in HipT$_{Ec}$) are also show since they potentially offer a stabilizing role.

The online version of this article includes the following figure supplement(s) for figure 6:

**Figure supplement 1.** Structural comparison between HipST$_{Ec}$ and HipST$_{Lp}$.

with two residues, Asp145 and Lys201. These have chemically similar counterparts in HipT$_{Ec}$ (Gln151 and Arg214) and thus might interact with the phosphorylated Ser59 when the Gly-rich loop is in its inward state, that is, in the absence of HipS. Interactions at both sites (Ser57 to the RxDR motif, Ser59 to Gln151/Arg214) would stabilize the loop in its inward state and possibly affect the function of HipS as antitoxin. Moreover, since interaction of the phosphoserine with two arginine side chains at Ser57 would be much stronger than to a Gln and an Arg, due to the ionic character of the side chains, this could also explain why P-Ser57 interaction is stronger and thus potentially also why HipS is unable to inactivate the S57D phosphomimetic HipT mutant (DIS).

Finally, we show that the serine residues in the HipT Gly-rich loop (Ser57 and Ser59) are important for toxicity but apparently not for auto-phosphorylation as we observe auto-phosphorylation in both the AIS and SIA variants of HipT. This is compatible with a model in which auto-phosphorylation serves to both control target (TrpS) phosphorylation and HipS binding, which could explain the need for two sites. The molecular mechanism controlling at which site auto-phosphorylation takes places is still not known, but the structures of HipBST AIS and HipBST SIA suggest that subtle changes in the loop can affect this (*Figure 3b*).

Together, our data support a complex network model for the interplay between the active site and the Gly-rich loop depicted in *Figure 6*. In this model, residues involved in both ATP binding and catalysis, the phosphoserine residues in the Gly-rich loop, and parts of HipS form a tightly interconnected network of interactions. The result of this complexity is that posttranslational modification and/or mutation of any single residue can have multiple downstream effects. For example, inactivation of the HipT active site through the D210A mutation appears to also affects the position of the Gly-rich loop as well as HipS binding, via direct interactions to the serine residues. Likewise, the position of the Gly-rich loop and binding of HipS are mutually interconnected so that HipS controls whether the loop can transition from an outward to an inward state and conversely, the serine residues and their phosphorylation state likely also control with what affinity HipS binds. This shifts the role of HipS from that of a classical antitoxin that simply blocks and inhibits the toxin active site to that of an allosteric enzymatic modulator. We do not yet understand whether auto-phosphorylation can take place with HipS bound or requires its release but note that our model is consistent with both these scenarios. The complexity of the network therefore makes it difficult to associate single roles to specific residues and/or modifications and requires that the full system is considered for each functional state.

# Materials and methods
## Strains and plasmids

Strains and plasmids used or generated in this study are listed in *Supplementary file 1b*, and DNA oligonucleotides in *Supplementary file 1c*. The construct expressing the HipT$_{S57A}$ variant was constructed by PCR mutagenesis (*Supplementary file 1c*) as previously described (*Vang Nielsen et al., 2019*). Plasmids were constructed as detailed below.

### pSVN68

The S57A mutation in *hipB-S-T*$_{His6}$ with optimized Shine-Dalgarno (SD) sequences for all three genes was created using pSVN61 (*Vang Nielsen et al., 2019*) and primers hipX S57A Fw and hipX S57A Rv in a site-directed plasmid mutagenesis PCR. The samples were pooled and digested with DpnI to remove the template plasmid before being transformed into *E. coli* DH5α.

### pSVN78

*hipB-S-T*$^{S57A}$$_{His6}$ with optimized SDs for all three genes was subcloned from pSVN68 by digesting with XbaI and XhoI, purifying the DNA fragment and ligating into pET-15b.

### pSVN88

The D233Q mutation in *hipB-S-T*$_{His6}$ with optimized SD sequences for all three genes was created using pSVN61 (*Vang Nielsen et al., 2019*) and primers hipX D233Q Fw and hipX D233Q Rv in a site-directed plasmid mutagenesis PCR. The samples were pooled and digested with DpnI to remove the template plasmid before being transformed into *E. coli* DH5α.

### pSVN96

*hipB-S-T*$^{D233Q}$$_{His6}$ with optimized SDs for all three genes was subcloned from pSVN88 by digesting with XbaI and XhoI, purifying the DNA fragment and ligating into pET-15b.

### pSVN141

pGH254::P$_{hipBST}$-*hipB'-laZ*, P$_{hipBST}$-*hipB'*, a fragment containing 224 bp upstream of the *hipB* gene plus the first 73 bp of the *hipB* gene was amplified from pKG127 using primers FP43 and RP32. The resulting PCR product was digested with EcoRI and BamHI and ligated into pGH254.

### pSVN178

*hipS*$^{W65A}$ was created by a two-step PCR reaction. Two fragments were amplified from pSVN109 using primers FP22 and *hipS* W65A Rv in one reaction and *hipS* W65A Fw and RP14 in the other. The resulting two PCR products were joined by a second round of PCR using both fragments as template DNA and primers FP22 and RP14. The final PCR product was digested with XhoI and EcoRI and ligated into pNDM220 (*Gotfredsen and Gerdes, 1998*).

## pSVN181

*hipB-S-T*<sup>D233Q</sup> with optimized SDs for all three genes was amplified from pSVN88 using primers FP21 and RP11. The resulting PCR product was digested with KpnI and HindIII and ligated into pBAD33 (*Guzman et al., 1995*).

## pSVN182

*hipB-S* with optimized SDs for both genes was amplified from pSVN87 using primers FP21 and RP14. The resulting PCR product was digested with KpnI and HindIII and ligated into pBAD33. **pSVN185**. *hipB-T*<sup>D233Q</sup> with optimized SDs for both genes was created by amplifying two PCR products from pSVN88: *hipB* using primers FP21 and RP42 and *hipT*<sup>D233Q</sup> using primers FP48 and RP11. The *hipB* fragment was digested with KpnI and PstI, while the *hipT*<sup>D233Q</sup> fragment was digested with PstI and HindIII. The two digested fragments were then ligated into pBAD33 using the KpnI and HindIII restriction sites.

## pSVN188

*hipS-T*<sup>D233Q</sup> with optimized SDs for both genes was amplified from pSVN87 using primers FP46 and RP11. The resulting PCR product was digested with KpnI and HindIII and ligated into pBAD33. **pSVN189**. *hipB* with optimized SD was amplified from pSVN87 using primers FP21 and RP15. The resulting PCR product was digested with KpnI and HindIII and ligated into pBAD33. **pSVN190**. *hipS* with optimized SD was amplified from pSVN87 using primers FP46 and RP14. The resulting PCR product was digested with KpnI and HindIII and ligated into pBAD33. **pSVN193**. *hipT*<sup>D233Q</sup> with optimized SD was amplified from pSVN88 using primers FP47 and RP11. The resulting PCR product was digested with KpnI and HindIII and ligated into pBAD33.

## pSVN194

*hipT*<sup>S57D</sup> with start codon GTG was created by a two-step PCR reaction. Two fragments were amplified from pKG127 using primers FP1(GTG) and HipT S57D Rv in one reaction and HipT S57D Fw and RP1 in the other. The resulting two PCR products were joined by a second round of PCR using both fragments as template DNA and primers FP1(GTG) and RP1. The final PCR product was digested with SalI and SphI and ligated into pBAD33.

## pSVN195

*hipT*<sup>S59D</sup> with start codon GTG was created by a two-step PCR reaction. Two fragments were amplified from pKG127 using primers FP1(GTG) and HipT S59D Rv in one reaction and HipT S59D Fw and RP1 in the other. The resulting two PCR products were joined by a second round of PCR using both fragments as template DNA and primers FP1(GTG) and RP1. The final PCR product was digested with SalI and SphI and ligated into pBAD33.

## pSVN199

*hipT*<sup>S57A</sup> with start codon GTG was amplified from pSVN78 using primers FP1(GTG) and RP1. The resulting PCR product was digested with SalI and SphI and ligated into pBAD33.

## pSVN201

*hipT*<sup>S59A</sup> with start codon GTG was created by a two-step PCR reaction. Two fragments were amplified from pKG127 using primers FP1(GTG) and HipT S59A Rv in one reaction and HipT S59A Fw and RP1 in the other. The resulting two PCR products were joined by a second round of PCR using both fragments as template DNA and primers FP1(GTG) and RP1. The final PCR product was digested with SalI and SphI and ligated into pBAD33.

## pSNN1

*hipT*<sup>S57A</sup> with optimized SD sequence was amplified from pSVN78 using primers hipBS del Fw and hipBS del Rv. This resulted in the deletion of *hipB* and *hipS*.

### pSNN2

*hipT*$^{S57A+D210A}$ with optimized SD sequence was amplified from pSNN1 using primers hipT D210A Fw and hipT D210A Rv.

### pMME3

*hipBST*$^{S59A}$ with optimized SD sequence was amplified from pSVN78 using primers hipT S57S59A Fw and hipT S57S59A Rv to introduce the mutations A57S and S59A.

### pRBS1

*hipB-S-T*$^{D210A}_{His6}$ in pET-15b was created using the Q5 Site-Directed Mutagenesis kit (NEB) according to the procedures from the manufacturer, using pSVN78 as template and primer Q5 HipT D210A Fw and Q5 HipT D210A Rv.

### pRBS2

*hipB-S-T*$^{S57D,D210A}_{His6}$ in pET-15b was created using Q5 Site-Directed Mutagenesis kit (NEB) according to the procedures from the manufacturer, using pRBS1 as template and primer Q5 HipT S57D Fw and Q5 HipT S57D Rv.

### pRBS3

*hipB-S-T*$^{S59D,D210A}_{His6}$ in pET-15b was created using Q5 Site-Directed Mutagenesis kit (NEB) according to the procedures from the manufacturer, using pRBS1 as template and primer Q5 HipT S59D Fw and Q5 HipT S59D Rv.

## Protein purification and structure determination

Expression of the *E. coli* O127:H6 HipBST complex was done by introduction of the constructs pSVN78 (HipBST$^{S57A}$), pSVN96 (HipBST$^{D233Q}$), and pMME3 (HipBST$^{S59A}$), into *E. coli* BL21 DE3. All constructs encode HipB, HipS, and C-terminal hexa-histidine tagged version of HipT, with genes separated and including individual, optimized SD sequences. All constructs contained an isopropyl-D-1 thio-galactopyranoside (IPTG)-inducible promoter and ampicillin resistance gene for selection. For each construct, 2 l cultures of *E. coli* BL21 DE3 grown in LB medium to a cell density of $OD_{600} = 0.6$ were induced with a final concentration of 1 mM IPTG and left to express overnight at 20°C with vigorous shaking. Cell pellets were resuspended in lysis buffer (50 mM Tris–HCl, pH 7.5, 300 mM NaCl, 20 mM imidazole, 5% glycerol, 5 mM beta-mercaptoethanol (BME), 1 mM phenylmethylsulfonyl fluoride (PMSF)) and lysed by sonication. The lysate was cleared by centrifugation and applied to a 5-ml HisTrap HP column (Cytiva) equilibrated in lysis buffer and subsequently washed in wash buffer (50 mM Tris–HCl, pH 7.5, 300 mM NaCl, 40 mM imidazole, 5 mM BME), before eluting with 50 mM Tris–HCl, pH 7.5, 300 mM NaCl, 400 mM imidazole, 5 mM BME. The eluate was applied directly to a 5-ml Heparin HF column equilibrated in 70% Buffer A (50 mM Tris–HCl, pH 7.5, 5 mM BME) and 30% Buffer B (50 mM Tris–HCl, pH 7.5, 1 M NaCl, 5 mM BME) attached to an ÄKTA Pure system (Cytiva). This step separated the population into fully formed HipBST complexes that bound to the column, and various partial complexes that were washed off. Final separation was achieved following concentrating the Heparin elution to approximately 8 mg/ml using a 30 kDa MWCO Vivaspin filter (Sartorius), by applying the sample to a Superdex 200 10/300 GL (Cytiva) column equilibrated in gel filtration buffer (20 mM Tris–HCl, pH 7.5, 300 mM NaCl, 5 mM BME). Finally, the sample was concentrated to a protein concentration of 5 mg/ml by spin filtration before crystallization. Crystals of HipBST mutants grew as clusters of thin plates in 2 µl drops containing a 1:1 ratio of protein to the crystallization buffer, which consisted of 0.1 M Bicine, pH 9, 8% 2-Methyl-2,4-pentanediol (MPD), set up against a reservoir of 200 µl of the same condition. Crystals were cryoprotected in crystallization buffer supplemented with 25% MPD before freezing in liquid $N_2$.

Data collection for HipBST$^{S57A}$ was carried out at the P14 beamline at EMBL/DESY, Hamburg and for HipBST$^{S59A}$ and HipBST$^{D233Q}$ at the BioMAX beamline at MAX IV in Lund, Sweden. For HipBST$^{S59A}$ and HipBST$^{D233Q}$, 7200 images were collected with an oscillation of 0.1° and a transmission of 100% while for HipBST$^{S57A}$, 3600 images were collected with an oscillation of 0.1° and a transmission of 70%. All data were processed in XDS (*Kabsch, 2010*), using the CC½ value after scaling to set the diffraction limit (*Karplus and Diederichs, 2012*). The space group was confirmed using Pointless (*Evans,*

*2006*), and all structures were determined using molecular replacement with Phaser (*McCoy et al., 2007*) from the CCP4 suite (*Winn et al., 2011*). For HipBST[S57A], a heavily truncated structure of *E. coli* HipA was used as search model, while for HipBST[S59A] and HipBST[D233Q], the preliminary structure of HipBST[S57A] was used. Iterative refinement was carried out in Buster (*Smart et al., 2012*) using one big cycle of 20 small cycles of refinement during model building, and 5 big cycles of 100 small cycles of refinement for the subsequent polishing. Non-crystallographic symmetry (NCS) restraints were used throughout since the ASU contained a dimer, but some refinement rounds without NCS were also included to allow the model to adapt to differences between the molecules in the ASU. Automatic water placement was performed in Buster and water molecules were manually pruned by inspecting the electron density. Model building was performed in Coot (*Emsley et al., 2010*). All structures were validated by the MolProbity server, and Rama-*Z* scores as calculated by Phenix (*Sobolev et al., 2020*). Final $R_{work}/R_{free}$ for HipBST[D233Q], HipBST[S57A], and HipBST[S59A], were 0.19/0.23, 0.21/0.24, and 0.20/0.24, respectively.

## Spot assays and growth curves

Cultures of relevant strains of *E. coli* were grown in liquid YT medium or MOPS minimal medium including 0.2% glucose at 37°C with shaking at 160 rpm. YT agar plates were used as solid medium and were incubated at 37°C for approximately 16 hr. For selection, media was supplemented with 25 µg/ml chloramphenicol, 30 µg/ml ampicillin, and/or 25 µg/ml kanamycin. For spot assays, *E. coli* cells were grown as overnight cultures, diluted to obtain identical $OD_{600}$, centrifuged at 5000 rpm for 5 min, washed in phosphate-buffered saline, and serially diluted before being spotted onto YT agar plates containing the indicated amount of inducer or repressor. Gene expression from plasmids carrying the pBAD promoter was induced by a final concentration of 0.2% arabinose and repressed by 0.2% glucose. Gene expression from plasmids carrying the $P_{A1/O4/O3}$ promoter was induced by a final concentration of 200 or 500 µM IPTG, as indicated. Growth experiments were done in YT medium with addition of relevant antibiotics, diluted from overnight cultures and grown exponentially for at least 4 hr until a constant doubling time was observed. At $OD_{600}$ ~0.2, 0.2% arabinose was added to induce expression of wild-type HipT or auto-phosphorylation mutants. After another 1.5 hr, 200 µM IPTG was added to induce expression of HipS. For each repetition, an independent colony from the strain was used to start separate cultures. No outliers were rejected.

## Phos-tag gel

15% Phos-tag acrylamide gels (Wako) were cast according to the manufacturer's guidelines, except that 100 µM Phos-tag acrylamide was added to ensure proper separation between phosphorylated and unphosphorylated HipT. The unphosphorylated inactive HipT[S57A+D210A] control was expressed from pSNN2. The gel was run at 4°C until the loading dye reached the bottom of the gel and visualized using standard Coomassie Blue staining as for normal SDS–PAGE gels.

## SAXS measurements and analysis

HipBST[D210A], HipBST[S57D,D210A], and HipBST[S59D,D210A] were expressed from pRBS1, pRBS2, and pRBS3, and purified as described above. SAXS measurements were performed using an optimized NanoSTAR instrument (Bruker AXS), which uses a high brilliance Ga metal-jet X-ray source (Excillum), special long optics, a two-pinhole collimation with a custom 'scatterless' slit pinhole, and a VÅNTEC-2000 (Bruker AXS) microgap 2D gas proportional detector (*Lyngsø and Pedersen, 2021*). Samples and buffer standards were measured in the same flow-through quartz capillary, and the scattering from the buffer was subtracted before the data were converted to absolute scale using the scattering from water. SAXS intensity data, $I(q)$, were analysed as a function of the modulus of the scattering vector $q = 4\pi sin(\theta)/\lambda$, where $2\theta$ is the scattering angle and $\lambda$ is the wavelength of the X-rays. Guinier analysis, giving radius of gyration $R_g$ and the forward scattering $I(0)$, and an indirect Fourier transformation were then calculated (*Glatter, 1977*; *Pedersen et al., 1994*), giving the same parameters and also the pair distance distribution function $p(r)$ and the maximum diameter $D_{max}$ of the particles. The predicted molar mass was then calculated by $M = I(0) N_A / (c\Delta\rho_m^2)$, where $N_A$ is Avogadro's number, $c$ is the mass concentration, and $\Delta\rho_m = 2.00 \times 10^{10} cmg^{-1}$ is the typical excess scattering length per unit mass for proteins. The data were also plotted in a normalized Kratky plot of $(qR_g)^2 I(q)/I(0)$ versus $qR_g$ to assess compactness and check for contributions from random coil parts. For structural

modelling, a AlpheFold2-predicted structure for N-terminus of HipB was added to initially have a molecular model with the correct molecular mass. The predicted SAXS curve of the model was then calculated as described in *Steiner et al., 2018* where the hydration layer is described by dummy atoms. the Debye equation is used (*Debye, 1915*) using an average form factor for all non-hydrogen atoms:

$$I\left(q\right) = cM\Delta\rho_m^2 P\left(q\right) = cM\Delta\rho_m^2\exp\left(-q^2\sigma^2\right)\sum_{i,j}b_ib_j\frac{\sin\left(qd_{ij}\right)}{qd_{ij}}$$

which defines the form factor $P(q)$, and where $\sigma$ = 1.0 Å, $b_i$ is, respectively, equal to the average excess scattering length of a non-hydrogen atoms for the protein and equal to an average excess scattering length for a hydration dummy atom. The parameter $d_{ij}$ is the distance between the $i$th and the $j$th atoms. Poor agreement was observed for all the data sets, so to improve the modelling, rigid-body refinement of the structure was performed using three bodies for HipB (residues 1–31, 32–43, and 44–107, respectively), one body for the HipS, and three bodies for HipT (residues 2–59, 60–169, and 170–331, respectively). Distance restraints were then added at the domain boundaries with additional restraints between the domains (Glu A107–Val D101, Pro B55–Gly C154, Trp B65–Gly C60, Gly D100–Aap F188, Leu D6–Arg F291, Val C147–Gly B94) based on analysis of the crystal structure, and finally a restraint of excluded volume was added. The Guinier and IFT analysis indicated some oligomerization in the samples, and therefore, a structure factor $S(q) = 1 + (N − 1)$ sin($qD$)/$qD$ (*Larsen et al., 2020*) for a dimer was included. In the expression, where $N$ is the average of the number of proteins in the oligomer, and $D$ is the distance between the two copies in the dimer. The structure factor was included in the decoupling approximation (*Kotlarchyk and Chen, 1983*):

$$I\left(q\right) / \left(cM\Delta\rho_m^2\right) = P\left(q\right) + \frac{\langle A\left(q\right)\rangle^2}{P\left(q\right)}\left[S\left(q\right) - 1\right]$$

with corresponding amplitude,

$$\langle A\left(q\right)\rangle = \exp\left(\frac{-q^2\sigma^2}{2}\right)\sum_i b_i\frac{\sin\left(qd_{i,CM}\right)}{qd_{i,CM}}$$

where $d_{i,CM}$ is the distance of the $i$th atom from the scattering centre of mass. During the optimization of the structure, the two parameters of the structure factor, $N$ and $D$, were also varied.

## Phylogenetic analysis

The previously identified set of 48 HipT orthologues (*Gerdes et al., 2021*) was used for sequence alignment by Clustal Omega (*Sievers and Higgins, 2018*) at https://www.ebi.ac.uk and imported into Jalview (*Waterhouse et al., 2009*). The phylogenetic tree was visualized using iTOL (*Letunic and Bork, 2019*). Reconstruction of the phylogenetic tree was accomplished using IQ-TREE that uses the Maximum Likelihood approach and Ultrafast bootstrapping via the CIPRES module in Genious Prime (*Minh et al., 2020*).

## Acknowledgements

The authors are indebted to the beamline staff at P14 in EMBL Hamburg, and BioMAX in MaxIV Lund for help during data collection. This project was funded by grants from the Novo Nordisk Foundation (NNF18OC0030646 to DEB) and Danish Natural Research Foundation's Centre of Excellence for Bacterial Stress Response and Persistence (grant number DNRF120).

## Additional information

### Funding

| Funder | Grant reference number | Author |
|---|---|---|
| Novo Nordisk Foundation | NNF18OC0030646 | René L Bærentsen |
| Danmarks Grundforskningsfond | DNRF120 | René L Bærentsen |

The funders had no role in study design, data collection, and interpretation, or the decision to submit the work for publication.

### Author contributions

René L Bærentsen, Stine V Nielsen, Conceptualization, Data curation, Formal analysis, Validation, Investigation, Visualization, Methodology, Writing – original draft, Writing – review and editing; Ragnhild B Skjerning, Visualization, Methodology, Project administration, Writing – review and editing; Jeppe Lyngsø, Data curation, Formal analysis, Visualization, Methodology, Writing – review and editing; Francesco Bisiak, Formal analysis, Validation, Investigation, Visualization, Methodology, Writing – original draft; Jan Skov Pedersen, Conceptualization, Resources, Data curation, Software, Formal analysis, Supervision, Funding acquisition, Validation, Investigation, Visualization, Methodology, Writing – original draft, Project administration, Writing – review and editing; Kenn Gerdes, Conceptualization, Supervision, Funding acquisition, Investigation, Writing – original draft, Project administration, Writing – review and editing; Michael A Sørensen, Ditlev E Brodersen, Conceptualization, Resources, Data curation, Formal analysis, Supervision, Funding acquisition, Validation, Investigation, Visualization, Writing – original draft, Project administration, Writing – review and editing

### Author ORCIDs

René L Bærentsen (ID) http://orcid.org/0000-0002-6460-4845
Ragnhild B Skjerning (ID) http://orcid.org/0000-0003-3886-8086
Jeppe Lyngsø (ID) http://orcid.org/0000-0002-6301-1300
Francesco Bisiak (ID) http://orcid.org/0000-0001-5375-6264
Michael A Sørensen (ID) https://orcid.org/0000-0001-8931-2999
Ditlev E Brodersen (ID) https://orcid.org/0000-0002-5413-4667

Reviewer #1 (Public Review): https://doi.org/10.7554/eLife.90400.3.sa1
Reviewer #2 (Public Review): https://doi.org/10.7554/eLife.90400.3.sa2
Author Response https://doi.org/10.7554/eLife.90400.3.sa3

# Additional files

### Supplementary files
• Supplementary file 1. (a) Crystallographic data statistics. Crystallographic data collection (upper part) and refinement (lower part) statistics for the HipBST$^{D233Q}$, HipBST$^{S57A}$, and HipBST$^{S59A}$ structures. *Numbers in parentheses refer to the outermost resolution shell. (b) Bacterial strains and plasmids. List of bacterial strains and plasmids either prepared as part of this work or with the given reference. SD, Shine-Dalgarno sequence. (c) Oligonucleotides and primers. List of oligonucleotides and primers used in this work, 5'–3' sequences.

• MDAR checklist

### Data availability
Diffraction data and structures have been deposited in PDB with accession IDs 7AB3 (HipBST SIA), 7AB4 (HipBST AIS), and 7AB5 (HipBST$^{D233Q}$).

The following datasets were generated:

| Author(s) | Year | Dataset title | Dataset URL | Database and Identifier |
|---|---|---|---|---|
| Baerentsen RL, Brodersen DE | 2022 | Crystal structure of the *Escherichia coli* toxin-antitoxin system HipBST (HipT S57A) | https://www.rcsb.org/structure/7AB3 | RCSB Protein Data Bank, 7AB3 |
| Baerentsen RL, Brodersen DE | 2022 | Crystal structure of the *Escherichia coli* toxin-antitoxin system HipBST (HipT S59A) | https://www.rcsb.org/structure/7AB4 | RCSB Protein Data Bank, 7AB4 |
| Baerentsen RL, Brodersen DE | 2022 | Crystal structure of the *Escherichia coli* toxin-antitoxin system HipBST (HipT D233Q) | https://www.rcsb.org/structure/7AB5 | RCSB Protein Data Bank, 7AB5 |

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
