## [Editor Report · eLife assessment]

This **important** study presents an exhaustive structural analysis of a complete tripartite HipBST toxin-antitoxin system of the Enteropathogenic *E. coli* O127:H6, which represents a fascinating variation on the well-studied HipAB toxin-antitoxin system. The **convincing** data show that major features of the canonical HipAB system have been rerouted to form the tripartite HipBST, revealing a new mode of inhibition of a toxin kinase.

---

## [Referee Report · Reviewer #1 (Public Review)]

This work describes a structural analysis of the tripartite HipBST toxin-antitoxin (TA) system, which is related to the canonical two-component HipBA system composed of the HipA serine-threonine kinase toxin and the HipB antitoxin. The crystal structure of the kinase-inactive HipBST complex of the Enteropathogenic *E. coli* O127:H6 was solved and revealed that HipBST forms a hetero-hexameric complex composed of a dimer of HipBST heterotrimers that interact via the HipB subunit. The HipS antitoxin shows a structural resemblance to HipA N-terminal region and the HipT toxin represents to the core kinase domain of HipA, indicating that in HipBST the hipA toxin gene was likely split in two genes, namely hipS and hipT.

-The structure also reveals a conserved and essential Trp residue within the HipS antitoxin, which likely prevents the conserved "Gly-rich loop" of HipT from adopting an inward conformation needed for ATP binding. This work also shows that the regulating Gly-rich loop of the HipT toxin contains conserved phosphoserine residues essential for HipT toxicity that are key players within the HipT active site interacting network and which likely control antitoxin binding and/or activity.

Strengths:

The manuscript is well written and the experimental work well executed. It shows that major features of the classical two-component HipAB TA system have somehow been rerouted in the case of the tripartite HipBST. This includes the N-terminal domain of the HipA toxin, which now functions as bona fide antitoxin, and the partly relegated HipB antitoxin, which could only function as a transcription regulator. In addition, this work shows a new mode of inhibition of a kinase toxin and highlights the impact of the phosphorylation state of key toxin residues in controlling the activity of the antitoxin.

Weaknesses:

The authors have convincingly addressed the previously raised weaknesses in their revised version of the manuscript.

---

## [Referee Report · Reviewer #2 (Public Review)]

The work by Bærentsen et al., entitled "Structural basis for regulation of a tripartite toxin-antitoxin system by dual phosphorylation" deals with the structural aspects of the control of the hipBST TA operon, the role of auto-phosphorylation in the activation and neutralisation of the enzyme and the direct effects of HipS and HipB in neutralisation. This is a follow-up to the Vang Nielsen et al., and Gerdes et al., papers from the same authors on this very unique TA module, that brings forth a thorough and well written dissection of an unusually complex regulatory system.

---

## [Author Response]

The following is the authors’ response to the original reviews.

**Reviewer #1 (Public Review):**
This work describes a structural analysis of the tripartite HipBST toxin-antitoxin (TA) system, which is related to the canonical two-component HipBA system composed of the HipA serine-threonine kinase toxin and the HipB antitoxin. The crystal structure of the kinase-inactive HipBST complex of the Enteropathogenic *E. coli* O127:H6 was solved and revealed that HipBST forms a hetero-hexameric complex composed of a dimer of HipBST heterotrimers that interact via the HipB subunit. The HipS antitoxin shows a structural resemblance to HipA N-terminal region and the HipT toxin represents to the core kinase domain of HipA, indicating that in HipBST the hipA toxin gene was likely split in two genes, namely hipS and hipT.-The structure also reveals a conserved and essential Trp residue within the HipS antitoxin, which likely prevents the conserved "Gly-rich loop" of HipT from adopting an inward conformation needed for ATP binding. This work also shows that the regulating Gly-rich loop of the HipT toxin contains conserved phosphoserine residues essential for HipT toxicity that are key players within the HipT active site interacting network and which likely control antitoxin binding and/or activity.Strengths:The manuscript is well written and the experimental work well executed. It shows that major features of the classical two-component HipAB TA system have somehow been rerouted in the case of the tripartite HipBST. This includes the N-terminal domain of the HipA toxin, which now functions as bona fide antitoxin, and the partly relegated HipB antitoxin, which could only function as a transcription regulator. In addition, this work shows a new mode of inhibition of a kinase toxin and highlights the impact of the phosphorylation state of key toxin residues in controlling the activity of the antitoxin.Weaknesses:A major weakness of this work is the lack of data concerning the role of HipB, which likely does not act as an antitoxin. Does it act as a transcriptional regulator of the hipBST operon and to what extent both HipS and HipT contribute to such regulation? These are still open questions.

We thank the reviewer for their feedback and have included a supplementary figure (Figure 1 supplement 2) and accompanying text that shows the transcriptional role of HipB, and how HipS and HipT influence this regulatory effect.

In addition, there is no in-depth structural comparison between the structure of the HipBST solved in the work and the two recent structures of HipBST from Legionella. This is also a major weakness of this work.

A structural comparison to the recent structures from Legionella has now been included in the discussion, including Figure 6 supplement 1.

**Reviewer #2 (Public Review):**
The work by Bærentsen et al., entitled "Structural basis for regulation of a tripartite toxin-antitoxin system by dual phosphorylation" deals with the structural aspects of the control of the hipBST TA operon, the role of auto-phosphorylation in the activation and neutralisation of the enzyme and the direct effects of HipS and HipB in neutralisation. This is a follow-up to the Vang Nielsen et al., and Gerdes et al., papers from the same authors on this very unique TA module, that brings forth a thorough and well written dissection of an unusually complex regulatory system.This is a much improved manuscript, the paper is more focused and the message is now clear.
**Reviewer #1 (Recommendations For The Authors):**
My main recommendation would be to include an in-depth structural comparison between the structure of the HipBST solved in the work and the two recent structures of similar HipBST from Legionella.

We thank the reviewer and have included a new supplementary figure (Figure 6 supplement 1) and expanded the comparison in the discussion to accommodate this.

**Reviewer #2 (Recommendations For The Authors):**
So I only have some minor comments.1. The authors should accompany Fig.1 (a supplementary panel is sufficient) with a surface electrostatic representation of the complex to better illustrate the potential role of the complex in transcription auto-regulation.

We have included a new panel in Figure 1 supplement 3 to show the electrostatic surface of the DNA-binding domains of HipB of HipBST and HipBASo.

1. When the Gly-rich loop is first introduced, please provide from which residue to which residue the loop expands.

Corrected for both the first mention of the Gly-rich loop of HipA and HipT.

1. In Fig 2. The authors try to show how the interaction of the main helix of HipS with HipT is different in HipBST compared to HipAB. I think it would be helpful if these two panel show the surface of HipT and HipA coloured by electrostatics so that not only the differences in HipS become apparent, but also the local differences between both toxins.

We thank the reviewer for this excellent idea, and the electrostatics did in fact reveal that the region of the toxins are different. We have updated figure 2b to show this difference.

1. Fig. 4 Shows the experimental SAXS curves for the HipT D210Q variants SIS (blue), SID (red), and DIS (orange). In each case a black curve is fitted to the data (presumably the fitting of the model-derived scattering curve to the data). Could the authors clarify this in the figure?

We agree that this information is missing in the legend. The black curves are the fits for the models based on the crystal structure after rigid-body refinements and inclusion of a structure factor to account for oligomerization of the complexes. This is now included in the figure caption.

1. Also regarding the SAXS analysis, in the manuscript the authors state that all three models "gave good fits to the data" as assessed by the fitting χ2. These χ2 values should be explicit in the figure or the figure legend.

We thank the reviewer for this suggestion. The chi squared values for the best fits are now given in the text.

In addition, is the SAXS data (the parameters derived from the experimental scattering, including the MW) consistent with the lack of HipS from the complex? (it should be...).

This is a good point, however, the partial oligomerization (dimerization) of the complexes (heterohexamers) and the variation of the dimerization degree between samples prevent extraction of useful mass values from the I(0) determinations. Therefore, we decided not to give the values explicitly in the text but only state “…consistent with analysis of the forward scattering that revealed partial oligomerisation of the samples with an average mass corresponding to roughly a dimer of the HipBST heterohexamer.”

1. Please improve this sentence: "Moreover, since it has previously been shown that only the HipT Gly-rich loop never is observed in doubly phosphorylated form with both Ser57 and Ser59 modified simultaneously, it is unlikely that the effects are due to autophosphorylation of the remaining serine residue in either case (Vang Nielsen et al., 2019)."

Done